Optimizing speleological monitoring efforts: insights from long-term data for tropical iron caves

Trevelin Leonardo Carreira 1
Simões Matheus Henrique 2
http://orcid.org/0000-0001-9260-1441 Prous Xavier 2
Pietrobon Thadeu 2
Brandi Iuri Viana 2
http://orcid.org/0000-0002-2101-5282 Jaffé Rodolfo 1 r.jaffe@ib.usp.br
1 Biodiversity and Ecosystem Services, Instituto Tecnológico Vale , Belém, Pará , Brazil
2 Environmental Licensing and Speleology, Vale S.A. , Nova Lima, Minas Gerais , Brazil
Baird Donald
Electronic publication date: 2021 Apr 16
Publication date: 2021
Volume: 9
Electronic Location ID: e11271
Received 2021 Jan 4; Accepted 2021 Mar 23
Copyright: © 2021 Trevelin et al.
Copyright year: 2021
Copyright holder: Trevelin et al.
License: This is an open access article distributed under the terms of the Creative Commons Attribution License, which permits unrestricted use, distribution, reproduction and adaptation in any medium and for any purpose provided that it is properly attributed. For attribution, the original author(s), title, publication source (PeerJ) and either DOI or URL of the article must be cited.
License URL: https://creativecommons.org/licenses/by/4.0/

Keywords: Iron caves, Landscape ecology, Mining, Speleology, Subterranean communities, Troglobites

Funding: Instituto Tecnológico Vale. RJ Conselho Nacional de Desenvolvimento Científico e Tecnológico 301616/2017-5 Funding was provided by Instituto Tecnológico Vale. RJ received a research productivity grant from Conselho Nacional de Desenvolvimento Científico e Tecnológico (301616/2017-5). The funders had no role in study design, data collection and analysis, decision to publish, or preparation of the manuscript.

==============================
Understanding the factors underpinning species abundance patterns in space and time is essential to implement effective cave conservation actions. Yet, the methods employed to monitor cave biodiversity still lack standardization, and no quantitative assessment has yet tried to optimize the amount and type of information required to efficiently identify disturbances in cave ecosystems. Using a comprehensive monitoring dataset for tropical iron caves, comprising abundance measurements for 33 target taxa surveyed across 95 caves along four years, here we provide the first evidence-based recommendations to optimize monitoring programs seeking to follow target species abundance through time. We found that seasonality did not influence the ability to detect temporal abundance trends. However, in most species, abundance estimates assessed during the dry season resulted in a more accurate detection of temporal abundance trends, and at least three surveys were required to identify global temporal abundance trends. Finally, we identified a subset of species that could potentially serve as short-term disturbance indicators. Results suggest that iron cave monitoring programs implemented in our study region could focus sampling efforts in the dry season, where detectability of target species is higher, while assuring data collection for at least three years. More generally, our study reveals the importance of long-term cave monitoring programs for detecting possible disturbances in subterranean ecosystems, and for using the generated information to optimize future monitoring efforts.

Introduction

Quantifying long-term changes in abundance of cave-dwelling organisms and identifying indicator species, reflecting the health status of subterranean ecosystems, are among the fundamental research goals of modern subterranean conservation biology (Mammola et al., 2020)⁠. For instance, the lack of knowledge about the factors underpinning abundance patterns in space and time are among the main impediments to the effective protection of cave fauna (Cardoso et al., 2011)⁠. Long-term studies in caves are scarce (Di Russo et al., 1997; Salvidio et al., 2019)⁠, and most previous efforts assessing community-level responses have evaluated population dynamics (Bichuette & Trajano, 2003; Ferreira et al., 2005; Lunghi, 2018)⁠, ecological niches (Mammola & Isaia, 2016; Mammola, Piano & Isaia, 2016)⁠, or temporal and spatial variation (Tobin, Hutchins & Schwartz, 2013; Ferreira et al., 2015; Owen et al., 2016; Paixão, Ferreira & Paixão, 2017; Mammola & Isaia, 2018; Ferreira & Pellegrini, 2019; Pellegrini, Faria & Ferreira, 2020)⁠. Few studies have evaluated the influence of anthropogenic disturbance on cave biodiversity (Bernardi, Souza-Silva & Ferreira, 2010; Pellegrini & Lopes Ferreira, 2012; Faille, Bourdeau & Deharveng, 2015; Cajaiba, Cabral & Santos, 2016; Pellegrini et al., 2016; Jaffé et al., 2018).

Due to the unique characteristics of subterranean environments, an important fraction of cave fauna exhibits adaptations for life in these extreme environments (Pipan & Culver, 2013)⁠. Some of these species are obligate subterranean dwellers and often comprise narrow-range endemic and threatened species (Harvey, 2002)⁠, so stringent legislation has been put in place in some countries to protect them (Harvey et al., 2011; Culver & Pipan, 2014)⁠. In Brazil, companies executing projects that could potentially impact cave ecosystems are required by law to assess the extent of impacts and implement control, monitoring and/or compensation measures (CONAMA, 1986; Brasil, 2008; MMA/ICMBio, 2019)⁠. After environmental licenses are granted, some caves are included in long-term monitoring programs, ultimately seeking to detect possible disturbances on subterranean fauna. These studies generate comprehensive biological databases containing valuable information for numerous caves sampled over long periods of time (Jaffé et al., 2016, 2018; Trevelin et al., 2019)⁠. However, although many recommendations have been made to monitor cave biodiversity (Eberhard, 2001; National Park Service, 2015; Culver & Sket, 2016)⁠, methods still lack standardization, and no quantitative assessment has yet tried to optimize the amount and type of information required to efficiently identify disturbances in cave ecosystems. This is nevertheless essential to design systematic, repeatable, and intensive surveys of cave-dwelling organisms, allowing the formulation of evidence-based management decisions (Wynne et al., 2018, 2019)⁠.

In Brazil, most cave monitoring programs have focused on assessing temporal changes in relative abundance in a set of selected species (ATIVO AMBIENTAL, 2019; BRANDT, 2019)⁠. However, the temporal frequency of field surveys, the impact of seasonal fluctuations in population size, and the sample sizes needed to detect temporal changes in population abundance, are yet to be systematically assessed. Moreover, the selection of species surveyed in these monitoring programs is not based on their usefulness as disturbance bio-indicators. Here we aim to fill these gaps, taking advantage of a comprehensive cave monitoring dataset containing abundance measurements for target taxa surveyed across iron caves along four years.

Material & methods

Study area

The study was performed in the Serra dos Carajás region, southeast of the state of Pará, in the Brazilian Amazon. This region is within the limits of the Floresta Nacional de Carajás, a protected area of 400,000 ha allowing sustainable use. The caves analyzed in this study are located in two highlands known as Serra Norte and Serra Sul (Fig. 1). These two regions harbor banded ironstone formations known as cangas, unique campo rupestre ecosystems resembling mountain savannas (Zappi et al., 2019)⁠, and one of the world’s largest deposits of iron ore (Poveromo, 1999)⁠.

Figure 1 Location of the study region (upper left corner) and a detail of the study area showing the spatial distribution of the caves included in our analyses (N = 95), colored by the number of surveys performed in each.

While the hillshade layer was constructed using a digital elevation model (SRTM, 1 arc-second) from USGS Earth Explorer, the land use classification shapefile was obtained from Souza-Filho et al. (2019). Coordinates are shown in decimal degrees.

Database

We used data generated by independent environmental consulting companies, so our study did not involve any field work. Vale S. A., a mining company, began operations in the region more than two decades ago (Souza-Filho et al., 2019)⁠, and has conducted numerous caves surveys over the last years as part of a large monitoring program related to environmental licensing processes. All surveys where authorized by Instituto Brasileiro do Meio Ambiente e dos Recursos Naturais Renováveis (IBAMA), under licenses ABIO 455/2014 (Projeto Ferra Carajás S11D n° 02001.000711/2009-46) and ABIO 639/2015 (Projeto Ferro Serra Norte–Estudo Global das Ampliações das minas N4 e N5 n° 02001.002197/2002-15). We compiled the data generated in these surveys to collect information from 33 target taxa across 95 caves, surveyed between August 2015 and September 2019. The selection of species included in these monitoring programs was based on the following criteria, as stated in environmental assessment reports (ATIVO AMBIENTAL, 2019; BRANDT, 2019)⁠: large body size and easy to identify in the field, abundant and showing a wide distribution range, resolved taxonomic classification (at least to the morpho-species level), and short life cycles allowing the rapid detection of changes in population dynamics (see Table 1 for the full list of target taxa and their ecological classification). All the selected species were actively surveyed during each field trip, so absences represent true absences rather than missing data. In each cave, the absolute abundance of each target taxa was quantified at least once during the rainy and the dry season, and sometimes multiple times in one year. Sampling was performed through an active visual search throughout the caves, aiming to cover all available micro-habitats (spaces under rocks, small cracks, moist soil, etc.) and organic deposits (litter, logs, carcasses, guano, etc.). Animals were collected with the aid of tweezers and brushes, and all individuals found in each cave were counted to estimate abundance per species, as performed in other studies (Silva, Martins & Ferreira, 2011; Ferreira et al., 2015; De Bento et al., 2016; Pellegrini & Ferreira, 2016; Paixão, Ferreira & Paixão, 2017; Ferreira & Pellegrini, 2019; Souza-Silva, Iniesta & Ferreira, 2020)⁠.

Table 1 List of surveyed taxa and their ecological classification.

Class	Order	Family	Species	Ecological Classification	
Malacostraca	Isopoda	Scleropactidae	Circoniscus carajasensis Campos-Filho & Araujo, 2011	Troglobiont	
Amphibia	Anura	Craugastoridae	Pristimantis cf. fenestratus (Steindachner, 1864)	Trogloxene	
		Leptodactylidae	Leptodactylus pentadactylus (Laurenti, 1768)	Accidental	
Arachnida	Amblypygi	Phrynidae	Heterophrynus longicornis Butler, 1873	Troglophile	
		Charinidae	Charinus ferreus Giupponi & Miranda, 2016	Troglobiont	
	Araneae	Araneidae	Alpaida sp.1	Troglophile	
		Pholcidae	Mesabolivar spp.	Troglophile	
		Prodidomidae	Prodidomidae sp.	Troglobiont	
		Salticidade	Astieae sp.1	Troglophile	
		Scytodidae	Scytodes eleonorae Rheims & Brescovit, 2001	Troglophile	
		Theraphosidae	Theraphosidae	Troglophile	
		Theridiosomatidade	Plato spp.	Troglophile	
	Opiliones	Cosmetidae	Roquettea singularis Mello-Leitão, 1931	Troglophile	
			Roquettea sp.	Troglophile	
		Escadabiidae	Escadabiidae sp.1	Troglobiont	
			Escadabiidae sp.2	Troglobiont	
		Gagrellinae	Prionostemma sp.	Troglophile	
		Stygnidae	Protimesius sp.	Troglophile	
			Stygnidae sp.1	Troglophile	
Chilopoda	Scutigeromorpha	Pselliodidae	Sphendononema guildingii (Newport, 1845)	Troglophile	
Diplopoda	Glomeridesmida	Glomeridesmidae	Glomeridesmus cf. spelaeus Iniesta, Ferreira & Wesener, 2012	Troglobiont	
	Polydesmida	Chelodesmidae	Chelodesmidae sp.	Troglophile	
		Pyrgodesmidae	Pyrgodesmidae sp.1	Troglobiont	
	Spirostreptida	–	Spirostreptida sp.	Troglophile	
		Pseudonannolenidae	Pseudonannolene cf. spelaea Iniesta & Ferreira, 2013	Troglobiont	
Insecta	Coleoptera	Dytiscidae	Dytiscidae sp.1	Stygobiont	
	Hemiptera	Cydnidae	Cydninae sp.1	Troglophile	
		Reduviidae	Emesinae sp.	Troglophile	
	Lepidoptera	Erebidae	Latebraria sp.	Trogloxene	
	Orthoptera	Phalangopsidae	Phalangopsis ferratilis Junta, Castro-Souza & Ferreira, 2020	Troglophile	
			Uvaroviella sp.	Troglophile	
Mammalia	Rodentia	Cricetidae	Rhipidomys sp.	Undefined	
Reptilia	Squamata	Phyllodactylidae	Thecadactylus rapicauda (Houttuyn, 1782)	Undefined	

Environmental conditions and landscape metrics

External and internal environmental conditions were monitored during the entire period across caves. Monitored variables included the deviation in average bimonthly rainfall in relation to the expected from a 20-years series (in mm, retrieved from small weather stations located in nearby mines S11D e N4E), and mean internal temperature (°C) on the date of the surveys (retrieved from portable data loggers placed in the most distant location from cave entrances). We also recorded the Area (meters2) of each studied cave as an additional internal condition widely known to influence biodiversity patterns in these ecosystems (Jaffé et al., 2016, 2018)⁠. Using 30 m resolution land-cover maps from 2015 to 2019 (Souza et al., 2020)⁠, we then quantified a suit of landscape metrics, including the proportional amount of forest, canga and mining land covers surrounding caves, and topographic distance to the nearest mine (see details in Table S1). These were all calculated at two different spatial scales (circular buffers with 500 and 1,000 m radius), using the R packages landscapemetrics (Hesselbarth et al., 2019)⁠ and TopoDistance (Wang, 2020)⁠. Two of these metrics directly captured possible disturbance of subterranean environments that could account for changes in the abundance of the studied species: mining cover and distance to the nearest mine.

Assessing drivers of community composition across caves

Aiming to quantify how environment, cave and landscape variables influenced overall community composition, we ran a partial redundancy analysis (RDA) controlling for differences between both highlands (Serra Norte and Serra Sul), using the vegan package (Oksanen et al., 2019)⁠. The community composition matrix containing relative abundances for each taxa was used as response variable and predictor variables included year, season, microclimate and landscape metrics (Legendre & Legendre, 1998)⁠. The highland where caves were located was specified as a conditional variable on the model to control for the effect of cave´s geographical location. Microclimate and landscape variables were standardized, community composition was Hellinger-transformed, and permutation tests were used to assess significance of marginal effects (Legendre & Legendre, 1998)⁠.

Assessing the influence of seasonality on the detection of temporal abundance trends

One of the main goals of cave monitoring programs was to assess changes in species abundance over time, and thereby identify species with declining or increasing populations in a particular cave. To understand how seasonality influenced the detection of abundance trends over time, we ran linear models containing the total number of observed individuals as the response variable and the interaction between sampling date and season. If seasonality influences temporal abundance trends, we would expect to find significant interaction terms. No significant interactions, on the other hand, would indicate that the trends can be detected regardless of the season when the surveys where performed. To prevent overfitting, linear models were ran for taxa and caves represented by at least five surveys in each season (final sample size was 16 taxa and 50 caves). Given the large number of models we used the Benjamini & Hochberg approach to adjust p-values, employing the p.adjust function from the stats R package (R Development Core Team, 2020)⁠.

Assessing the influence of sampling effort on the detection of temporal abundance trends

Given the extensive field exposure of people and elevated costs associated with cave monitoring programs, it is important to quantify how the sampling effort influences the detection of temporal abundance trends. To do so we compared linear model coefficients of models fitted with the full dataset with those of models fitted with reduced datasets. We first split the data by season and ran linear models containing the total number of observed individuals as the response variable and sampling date as predictor. In these full models we included observations for all sampling dates, and excluded taxa and caves represented by less than three surveys per season. We then ran linear models on data subsets containing a reduced number of observations (ranging between two and the maximum number of sampling dates found in each cave and taxa). For each data subset containing a given number of observations (surveys) we performed ten random samplings without replacement, to ensure the sampling of different sampling dates. Finally, we compared coefficients from full models with those of subset models using root mean squared error (rmse), implemented through the rmse function from the Metrics R package (Hamner & Frasco, 2018)⁠. Lower values of rmse indicate more similar model coefficients.

Identifying disturbance indicator species

Given the life history variation between species and their different susceptibility to habitat disturbance, it is essential to identify indicator species that show a rapid response to disturbance in order to optimize monitoring programs. By focusing on these indicator species, monitoring programs could survey caves more efficiently, thereby making resources available to study more caves or other aspects of cave biodiversity requiring attention. Here we tried to identify disturbance indicator species by assessing the relationship between disturbance metrics and species abundance patterns. We first modeled patterns of relative abundance across all caves, using the function manyglm from the R package mvabund (Wang et al., 2012)⁠. It uses a multivariate generalized linear model (GLM) to make inferences by fitting separate GLMs to a common set of explanatory variables, and testing significance through resampling-based hypothesis testing. We ran negative binomial GLMs containing abundance as the response variable and sampling season nested in year, distance to mine and mining cover as predictor variables. Significance p-values were calculated using 999 resampling iterations via PIT trap resampling, adjusted for multiple testing using a step-down resampling procedure (Wang et al., 2012)⁠. We then used univariate coefficient estimates and significance for individual species, to identify specific responses to disturbance metrics (distance to mine and mining cover).

We then assessed the relationship between disturbance metrics and temporal trends in species abundance within each cave. To do so we ran linear models containing the total number of observed individuals as the response variable and sampling date as predictor, excluding taxa and caves represented by surveys spanning less than three years (some caves where surveyed multiple times in a single year but these where only included in this analysis if surveys spanned at least three different years). We used the model coefficients for each species at each cave, representing temporal abundance trends (positive coefficients showing an increase and negative coefficients a decrease in abundance through time), to run a second set of linear models regressing temporal abundance trends on disturbance metrics. These second set of models thus contained as response variable the model coefficients representing temporal abundance trends for each species at each cave, and distance to mine and mining cover (at different spatial and temporal scales) as predictors. To prevent overfitting we excluded species represented by less than ten coefficients (caves), and only constructed models containing a single predictor. We then ran likelihood-ratio tests, where we compared each model with a null model containing no predictors, and selected those predictor variables resulting in a significant increase in the model’s log-likelihood. Finally, we retrieved and plotted coefficients and p-values for these best-fitting models. All data and R scripts are available in GitHub (https://github.com/rojaff/cave_monitoring).

Results

Overall community composition was weakly influenced by seasonality, cave size, environmental conditions, and the composition and configuration of landscapes surrounding caves, as more than 87% of variance in community composition remained unexplained by these factors (Table 2).

Table 2 Summary of partial redundancy analyses (RDA).

Variable	Df	Variance	F	Pr(>F)	
Season nested in year	1	0.0018	4.408	0.001***	
Canga cover	1	0.0005	1.164	0.285	
Forest cover	1	0.0008	2.093	0.055*	
Mining cover	1	0.0010	2.378	0.034*	
Distance to mine	1	0.0027	6.766	0.001***	
Area	1	0.0197	48.696	0.001***	
Temperature	1	0.0054	13.408	0.001***	
Dev Rainfall	1	0.0002	0.604	0.746	
Residual	671	0.2710			
Note:

The table shows F-statistics and p-values from permutation tests (adjusted r2 = 0.13). Significance is highlighted as * (p < 0.05), ** (p < 0.01) and *** (p < 0.001).

Seasonality did not influence the ability to detect species abundance trends over time, since the interaction effect between sampling date and season was not significant in any taxa nor cave (Fig. 2). Increasing the number of samples resulted in more similar model coefficients between full and subset models, and root mean squared errors usually stabilized after three surveys (Fig. 3). However, in most species the dry season datasets allowed a more accurate detection of temporal abundance trends, as revealed by lower root mean squared errors (Fig. 3).

Figure 2 Adjusted p-values for the interaction between sampling date and season across 16 taxa and 50 caves.

The Benjamini & Hochberg approach was used to adjust p-values and the red horizontal line shows the threshold value of 0.05 (values above this line represent cases where the interaction effect was not significant).

Figure 3 Root mean squared error (rmse) for model coefficients from full models and those of subset models containing reduced numbers of samples.

Lower values of rmse indicate more similar model coefficients (and a more reliable estimation of temporal abundance trends). For each data subset containing a given number of observations (surveys) we performed ten random samplings without replacement, to ensure the sampling of different sampling dates.

Whereas relative abundance was associated to at least one disturbance metric in 22 species (Fig. 4), temporal trends in abundance were found associated with disturbance metrics in only five species (Fig. 5). Overall, two taxa displayed consistent responses across effects, which makes them potential indicator species for cave monitoring programs: The troglobiont Charinus ferreus, which appeared negatively affected by disturbance, and a species belonging the Theraphosidae family, which seem to be favored by disturbance (Table 3).

Figure 4 Model coefficients and 95% confidence intervals for species showing significant associations between overall abundance and two disturbance metrics.

Figure 5 Model coefficients and 95% confidence intervals for species showing significant associations between temporal abundance trends and two disturbance metrics.

Table 3 Taxa displaying significant responses to disturbance metrics, considering overall abundance and temporal abundance trends.

Taxon	Abundance	Temporal abundance trend	Sampling	
Distance to mine	Mining cover	Distance to mine	Mining cover	
Charinus ferreus*	-	Negative	Positive	Negative	Dry	
Theraphosidae	Negative	Positive	-	Positive	Dry	
Uvaroviella sp.	Negative	–	Positive	Negative	Both	
Rhipidomys sp.	–	–	Positive	–	Dry	
Roquettea sp.	–	–	–	Positive	Rain	
Pyrgodesmidae sp.1*	–	Negative	–	–	Rain	
Spirostreptida sp.1	Positive	–	–	–	Rain	
Prodidomidae sp.*	Positive	Negative			Dry	
Escadabiidae sp.1*	Negative	Negative	–	–	Rain	
Escadabiidae sp.2*	Positive	–	–	–	Rain	
Leptodactylus pentadactylus	Negative	–	–	–	Both	
Pristimantis fenestratus	Negative	Positive	–	–	Dry	
Thecadactylus rapicauda	–	Negative	–	–	Dry	
Plato spp.	Negative	–	–	–	Dry	
Sphendononema guildingii	–	Positive	–	–	Dry	
Astieae sp.1	Negative	–	–	–	Dry	
Protimesius sp.	Negative	–	–	–	Rain	
Prionostemma sp.	Negative	Positive	–	–	Dry	
Stygnidae sp1	–	Positive	–	–	Dry	
Phalangopsis sp.1	–	Positive	–	–	Dry	
Notes:

* Troglobitic species.

Taxa showing consistent responses (highlighted in bold) are suggested as short-term disturbance indicators. The best sampling season (according to Fig. 3), is indicated for each taxa.

Discussion

By analyzing abundance measurements for 33 target taxa surveyed across 95 caves along four years, we found that overall community composition was weakly influenced by seasonality, cave size, environmental conditions, and the composition and configuration of landscapes surrounding caves. Furthermore, our results show that seasonality did not influence the ability to detect abundance trends over time. However, in most species, abundance estimates assessed during the dry season resulted in a more accurate detection of temporal abundance trends, and at least three surveys were required to identify global temporal abundance trends. Finally, we identified a subset of species that could potentially serve as short-term disturbance indicators, some showing consistent responses in different analyses.

Subterranean communities have been shown to be affected by seasonality, environmental conditions, cave characteristics, and the structure of surrounding landscapes (Simões, Souza-Silva & Ferreira, 2015; Pellegrini et al., 2016; De Bento et al., 2016; Mammola & Isaia, 2018; Salvidio et al., 2019; Pellegrini, Faria & Ferreira, 2020; Rabelo, Souza-Silva & Ferreira, 2020). However, our results reveal that overall community composition was only weakly influenced by these factors, as our model explained merely 13% of total variation in community composition (Table 2). In contrast, previous work have found that cave morphology, microclimate, cave depth, and sampling date explain up to 50% of the variation in community structure in limestone and marble caves (Tobin, Hutchins & Schwartz, 2013; Lunghi, Manenti & Ficetola, 2014)⁠. Our results thus suggest that other factors, not considered in our analyses, play an important role structuring subterranean communities of iron caves. Inter-specific interactions, for instance, are known to have a profound influences on community structure (Ferreira & Martins, 1999; Mammola, Piano & Isaia, 2016)⁠. Alternatively, biological samples collected in iron caves may not capture the dynamics of the entire subterranean habitat, comprised by a network of fissures and voids and traditionally referred to as Milieu Souterrain Superficiel (MSS) (Culver & Pipan, 2014; Mammola et al., 2016; Mammola, 2018)⁠. For instance, most of the surveyed caves were larger than 5 × 5 m (Fig. S1), so they did not represent suitable sampling sites for the MSS (Mammola et al., 2016)⁠.

Even though seasonality affected overall community composition, it did not influence the ability to detect species abundance trends over time. External climatic conditions are increasingly attenuated at higher cave depths (Tobin, Hutchins & Schwartz, 2013)⁠, so species occurring in the inner portions of caves appear to have life cycles decoupled from external seasons, whereas species inhabiting the outermost portions of caves seem to be more strongly affected by seasonality (Di Russo et al., 1997; Gunn, Hardwick & Wood, 2000; Bichuette & Trajano, 2003; Ferreira et al., 2015; Mammola, Piano & Isaia, 2016; Lunghi, 2018)⁠⁠. Recognizing the impact of seasonality on species detection, the current Brazilian legislation stipulates that cave biodiversity surveys need to comprise at least two sampling events, one during the dry and one during the rainy season (MMA, 2017)⁠. It is worth emphasizing that these sampling requirements targeted a more accurate estimation of species richness, but not the continuous monitoring of focus species in time. Two sampling events are likely insufficient to obtain reliable species richness estimates for highly diverse caves (Auler & Piló, 2015; Wynne et al., 2018)⁠, so some authors have argued for the estimation of optimal sample sizes based on species accumulation curves (Trajano & Bichuette, 2010; Trajano, 2013)⁠. Our results provide the first evidence-based recommendations to optimize sampling efforts of monitoring programs seeking to assess target species abundance through time. Specifically, our findings suggest that monitoring efforts aiming to detect changes in abundance through time do not need to sample during two different seasons each year (Fig. 2). Sampling efforts of such monitoring programs could thus be optimized by performing more focused surveys and by surveying a larger number of caves during the same period each year. Importantly, restricting sampling to a single season could substantially attenuate the negative impact of cave visitation by researchers on subterranean communities (Pellegrini & Ferreira, 2016; Pellegrini & Lopes Ferreira, 2012; Bernardi, Souza-Silva & Ferreira, 2010).

Although the composition and spatial distribution of subterranean communities can remain constant over periods of several years (Salvidio et al., 2019)⁠, our results suggest that sampling during at least three years is necessary to detect temporal changes in abundance patterns in most of our focus species (Fig. 3). We note that our dataset only spans a period of four years (although some caves were sampled multiple times during the same season/year), so it cannot capture longer temporal changes in abundance. We also caution that these results cannot be generalized to all subterranean fauna, as different life histories and generation times will ultimately determine how fast these organisms respond to disturbances (Ferreira, 2005; Mammola et al., 2016; Culver & Pipan, 2019)⁠. Sampling in different seasons did not influence the ability to detect general abundance trends over time, but the dry season datasets allowed a more accurate detection of temporal abundance trends in most species. These results suggest higher detection probabilities in the dry season for the subset of species where RMSE curves show a steeper decrease during the dry season (Fig. 3). Interestingly, this was the case for the troglobitic amblypygid Charinus ferreus, a species that is difficult to detect like other troglobionts (Wynne et al., 2018; Lunghi, 2018)⁠. Our results thus suggest that monitoring programs focusing on terrestrial subterranean fauna from our study region could concentrate sampling activity in the dry season, where most species seem to be easier to detect. Likewise, our findings highlight the importance of implementing long-term monitoring efforts spanning at least 3 years.

The concept of indicator species in ecosystem management relies on the idea of identifying taxa responsive to environmental change, that could inform policies, protocols, and best practices (Carignan & Villard, 2002)⁠. Such environmental indicators (McGeoch, 1998)⁠ seek to provide cost and time effective guidelines to address pressing conservation issues, such as those faced by large-scale mining projects (Sonter, Ali & Watson, 2018)⁠. Assessing the response of subterranean fauna to anthropogenic disturbance nevertheless requires access to long-term cave monitoring datasets, which are remarkably rare for tropical caves (McGeoch, 1998; Carignan & Villard, 2002; Mammola et al., 2020)⁠. Here we identified 20 taxa where overall abundance responded to cave disturbance, and five where temporal abundance trends where associated with disturbance. Only two taxa displayed consistent responses across effects, which makes them candidate indicator species for cave monitoring programs: Charinus ferreus and a species belonging the Theraphosidae family (Table 2). Both are arachnids, a group that was recently identified as biodiversity indicator for iron caves (Trevelin et al., 2019)⁠. Being a top predator restricted to cave ecosystems, the first species is a well-known troglobitic Amblypygi (De Lao Giupponi & De Miranda, 2016)⁠. Its strong and consistent response to disturbances (Figs. 4 and 5) suggest the species is associated with pristine and undisturbed ecosystems, which makes it an ideal disturbance indicator. Theraphosidae spiders, on the other hand, are sedentary sit-and-wait predators from the epigea, rarely occupying subterranean environments for reproduction or shelter (Fonseca-Ferreira, De Zampaulo & Guadanucci, 2017)⁠. Our results suggest that they apparently benefit from disturbance to opportunistically colonize caves, or alternatively, that disturbances in the surrounding external habitats are forcing them to look for shelter inside the caves. The species nevertheless awaits formal taxonomic description, which currently limits its usefulness as an indicator species.

Effect sizes of disturbance on overall abundance and temporal abundance trends where generally small, suggesting that some effects could have remained undetected because they would require sampling over longer time periods (Di Stefano, 2001; Legg & Nagy, 2006)⁠. For instance, the ability to detect trends in tropical bat population abundance was shown to be dependent on the duration of the monitoring efforts, and only long programs (>20 years) showed sufficient statistical power to reliably detect abundance trends (Meyer et al., 2010)⁠. This could explain why some of our focus species did not exhibit coherent responses across analyses, like the troglobionts Pyrgodesmidae sp. and Escadabiidae sp., or opportunistic colonizers like the anuran Leptodactylus pentadactylus or Pristimantis fenestratus. Although empirical evidence from long-term cave monitoring efforts focusing on invertebrates is scarce (Faille, Bourdeau & Deharveng, 2015; Cajaiba, Cabral & Santos, 2016; Owen et al., 2016)⁠, our results suggest that longer monitoring efforts are needed to detect disturbance responses in most cave-dwelling species.

Conclusions

Our study reveals the importance of long-term cave monitoring programs for detecting possible disturbances in subterranean ecosystems, and for using the generated information to optimize future monitoring efforts. Results show that iron cave monitoring programs implemented in our study region could focus sampling efforts in the dry season, where detectability of target species is higher, while assuring data collection for at least three years. Charinus ferreus was identified as the most promising short-term disturbance indicator species.

Supplemental Information

Supplemental Information 1 Environmental and landscape metrics assessed.

Click here for additional data file.

Supplemental Information 2 Frequency distribution of cave area for the 95 surveyed caves.

The red vertical line indicates an area of 25 m² (5x5m caves).

Click here for additional data file.

We thank Ativo Ambiental and Brandt Meio Ambiente for performing the speleological surveys that generated all the data used in this study, and David Culver and an anonymous reviewer for helping us improve our manuscript with their constructive criticism.

Additional Information and Declarations

Competing Interests

Author Contributions

Field Study Permissions

Data Availability

Rodolfo Jaffé is an Academic Editor for PeerJ. Matheus Simoes, Xavier Prous, Thadeu Pietrobon, and Iuri Viana Brandi are employees of Vale S.A.

Leonardo Carreira Trevelin conceived and designed the experiments, performed the experiments, analyzed the data, prepared figures and/or tables, authored or reviewed drafts of the paper, and approved the final draft.

Matheus Henrique Simões conceived and designed the experiments, performed the experiments, authored or reviewed drafts of the paper, and approved the final draft.

Xavier Prous performed the experiments, authored or reviewed drafts of the paper, and approved the final draft.

Thadeu Pietrobon performed the experiments, authored or reviewed drafts of the paper, and approved the final draft.

Iuri Viana Brandi conceived and designed the experiments, authored or reviewed drafts of the paper, and approved the final draft.

Rodolfo Jaffé conceived and designed the experiments, analyzed the data, prepared figures and/or tables, authored or reviewed drafts of the paper, and approved the final draft.

The following information was supplied relating to field study approvals (i.e., approving body and any reference numbers):

We did not collect specimens but employed survey data generated by environmental consulting companies. All surveys where authorized by Instituto Brasileiro do Meio Ambiente e dos Recursos Naturais Renováveis (IBAMA) [ABIO 455/2014 (Projeto Ferra Carajás S11D n° 02001.000711/2009-46) and ABIO 639/2015 (Projeto Ferro Serra Norte – Estudo Global das Ampliações das minas N4 e N5 n° 02001.002197/2002-15)].

The following information was supplied regarding data availability:

All data and R scripts are available in GitHub: https://github.com/rojaff/cave_monitoring

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
