# Peer review of "Optimizing speleological monitoring efforts: insights from long-term data for tropical iron caves"

_PeerJ, doi:10.7717/peerj.11271_

## Round 0.1 · original submission · Major Revisions

Reviewer 1 has indicated that there are several areas of the manuscript which need attention. In particular, the results section is extremely short, and requires greater development and elaboration. Also, you should take care to align the methods major sections with the results section. Reviewer 2 has fewer concerns, but notes that the issue of cave size/area might be usefully addressed in the discussion.

·

Basic reporting

I found this to be a very well written paper, and I didn't have my usually minor comments. With one exception, discussed below, the literature review is sufficient.
Iron ore caves are very interesting both from a biological and conservation point of view, and the authors do a good job conveying this.

Experimental design

I am not an expert on statistical modelling, but the authors persuaded me that they used appropriate tests and procedures.

Validity of the findings

The relatively low explanatory power (line 250) is disappointing, but I thought that perhaps one of the reasons is that much of the habitat is not caves per se but rather than MSS in the canga itself. Most iron ore caves, at least in the Iron Triangle where I have worked, are very small, and many don't even have a dark zone. Perhaps sampling in caves is not sampling the bulk of the species or populations. I am not saying that this appeal to the MSS invalidates the study but somehwere it is surely worth a mention.

Additional comments

I found that there was a fresh approach to monitoring and i think it is a significant contribution to the literature.

Reviewer 2 ·

Basic reporting

I did not complete the review of this paper. The paper seemed rather underdeveloped. There were also organizational issues, as well as several key references were overlooked. As such, I recommend a major revision. Once these issues are addressed, I'd be more than happy to review the paper again.

Experimental design

Major revision requested. Experimental design is not clearly written. They vacillate between stating they used existing data and collected field data. I wasn't sure which.

Validity of the findings

This section was surprisingly underdeveloped. They need to present their results using the same structure provided for their methods. In other words, they use apply the same subsections in the results there were presented in the methods. And thus fully develop their results using that approach.

Additional comments

Please find general comments as sticky notes on the PDF.

Annotated reviews are not available for download in order to protect the identity of reviewers who chose to remain anonymous.

---

## Round 0.2 · accepted · Accept

You have addressed reviewers' comments directly, and I have recommended acceptance of your paper.

·

Basic reporting

No comment

Experimental design

No comment

Validity of the findings

No Comment

Additional comments

My comments were addressed.